# Hierarchical Self-Assembly of Dipolar ZnO Nanoparticles and Microdroplets

**DOI:** 10.3390/mi13091522

**Published:** 2022-09-14

**Authors:** Najla Ghifari, Rachid Bennacer, Adil Chahboun, Abdel I. El Abed

**Affiliations:** 1Laboratoire Lumière Matière et Interfaces (LUMIN), UMR 9024, Ecole Normale Supérieure Paris Saclay, CentraleSupélec, CNRS, Université Paris-Saclay, 4 Avenue des Sciences, 91190 Gif-sur-Yvette, France; 2ENS Paris-Saclay, CNRS, LMPS, Université Paris-Saclay, 91190 Gif-sur-Yvette, France; 3Laboratoire des Couches Minces et Nanomatériaux (CMN), FST Tanger, Université Abdelmalek Essaadi, BP 416, Tangier 90000, Morocco

**Keywords:** droplets, microfluidics, meso-crystallization, ZnO nanoparticles, dipolar electric interaction, tip-streaming effect

## Abstract

In this work, we investigated the orientation and the polarization of ZnO nanoparticles, which serve as building blocks of highly monodisperse microspheres, using a droplet microfluidic-assisted synthesis method. We observe, for the first time, a square lattice organization of liquid microdroplets, in a steady state, at the oil/water interface. Such square organization reveals clearly a dipolar organization of ZnO nanoparticles at the surfaces of droplets at the early stage of ZnO nanocrystal aggregation and microsphere formation. We discuss different models of organization of ZnO nanoparticles and show that the well-known tip-streaming effect in droplets in microfluidics explains the reason for the obtained dipolar droplets. The square organization is illustrated and explained.

## 1. Introduction

Mesocrystals are a fascinating class of colloidal materials with mesoscopically self-organized structures [1]. They build up via the oriented self-assembling of individual nanosized building units, in such a way that the overall orientation of the resulting larger nano- and micro-particles is single-crystal-like [2,3,4,5,6,7,8,9,10,11,12]. Their fabrication relies on the more general concept of the so-called hierarchical self-assembling process, where small structural units (e.g., molecules, ions, nanocrystals) serve as building blocks for more complicated meso-structures, which may find potential applications in many scientific and technological domains, such as nano- and micro-technologies, pharmaceuticals, cosmetics, or material science. Their nucleation and growth rely on specific interactions between building blocks, such as hydrogen bonding, electrostatic interactions, hydrophobic interactions, or host–guest interactions [13].

The understanding and control of hierarchical self-assembly at the nano and micro scales is therefore of great importance. It has been proven, for instance, that the electrical charges or dipole moments carried by the building units play a critical role in this non-classical crystallization process. Busch and coworkers suggested that intrinsic electric fields may be responsible for the orientation and self-organization of the individual primary building blocks [14,15].

A particular emphasis has been placed, during the last few decades, on understanding the relation between the crystal structure and the polarity of oxide materials with remarkable features, such as piezo-electricity, second harmonic generation, or ferro-electricity. Zinc oxide (ZnO) has been particularly scrutinised due to the significant intrinsic dipole moment of its wurtzite crystal structure, which is inherent to its tetrahedral configuration, where each type of ion ($Zn^{2+}$ or $O^{2-}$) has four neighbouring ions of the other type, and vice versa. In other words, the tetrahedral coordination exhibits a sequence of positively charged $Zn^{2+}$ and negatively charged $O^{2-}$ polar planes within the c-axis direction, respectively contributing to two opposite faces of polarity (0001) and $\left(000\bar{1}\right)$ perpendicular to the c-axis [16,17,18,19,20], as illustrated in Figure 1. The stacking of ZnO nanoparticles along their c-axis is non-centrosymmetric, which leads to a pronounced spontaneous polarization and a large intrinsic dipole moment along the c-axis and increasing with the nanoparticles’ thickness [21]. Typical values of approximately 50 mC/m${}^{2}$ were reported in the literature for the spontaneous polarization of ZnO nanocrystals [22,23,24,25].

We have developed recently an original approach using droplet microfluidics, which permits the synthesis of highly monodisperse microcapsules with a shell thickness of a few hundred nanometers [26]. This approach, illustrated in Figure 2, brings a new and detailed insight into the self-assembling of nanoparticle building units and their organization in micro-sized meso-structures.

This approach enables not only smart control of the size and content of droplets, but enables us also to control the orientation and organization of the polar ZnO nanoparticle building units at the surfaces of droplets and resulting microcapsules. Such control may be achieved, for instance, by changing the interfacial tension at the droplet/oil interface, or the screening of electric interactions between ZnO nanoparticles by adding ions or charged molecules [26].

In this work, we investigate the organization of ZnO nanoparticle building units at the very early stage of ZnO droplet formation in the microfluidic channel and after their collection and organization at the oil/air interface in a Petri dish. We observe, for the first time, a square lattice organization of liquid microdroplets, which reveals, unambiguously, the dipolar self-organization of ZnO nanocrystals at the surfaces of microdroplets, at the early stage of droplet formation. We suggest that the rise of a global electric dipole moment of droplets is triggered intrinsically by the microfluidic flow and the well-known tip-streaming effect in droplet microfluidics [27,28,29].

## 2. Methodology

A detailed description of the fabrication process of the used microfluidic devices and the synthesis of ZnO microdroplets and microcapsules was given with more details in a previous study [26].

Briefly, we used droplet generators with a flow-focusing geometry, which were fabricated using a standard soft lithography technique [30] and polydimethylsiloxane polymer (PDMS). We first used an SU-8 2025 photoresist for the manufacture of a master mold; then, the PDMS microfluidic device block was prepared by mixing a prepolymer (Sylgard 184 ™ from Neyco-France) and a curing agent (10% wt.). The mixture was degassed using a vacuum pump at room temperature, and then poured over the master mold, placed into a Petri dish, and thermally cured at 75 ${}^{{}^{\circ}}$C for 2 h. Finally, the fabricated PDMS replica was easily removed from the master mold and punched to create the inputs and outputs of the microfluidic device.

For the synthesis of ZnO nanoparticles, we dissolved 0.6 g of zinc acetate in 5 mL of methanol $\;(CH_{3}OH)$, which corresponds to a concentration of 0.55 M of zinc oxide. The formation mechanism of ZnO nanoparticles from zinc acetate involves a series of steps. First, the high dielectric constant of methanol contributes to the dissolution of zinc acetate dehydrate. Its dehydration results in the formation of anhydrous zinc acetate and water and acetate, which will later turn into acetic acid. Then, the solution is stirred for 1 h at 60 ${}^{{}^{\circ}}$C under magnetic stirring to ensure homogeneous mixing and obtain a transparent solution. A chemical reaction between the species present in the slightly basic solution takes place, leading to the precipitation of zinc hydroxide, $\;Zn{\left(OH\right)}_{2}$, and ZnO [31]. Ultimately, zinc hydroxide is converted to zinc oxide. The formation mechanism of zinc oxide is achieved according to the following chemical reactions:$$\begin{array}{c} Zn{\left(CH_{3}COO\right)}^{2-}⟶Zn^{2+}+2CH_{3}COO^{-}\\ Zn^{2+}+2OH^{-}⟶Zn{\left(OH\right)}_{2}\\ Zn{\left(OH\right)}_{2}⟶ZnO+H_{2}O \end{array}$$

The as-synthesized ZnO nanoparticle dispersion was then used to produce highly monodisperse droplets of ZnO nanoparticles, as illustrated in Figure 2a. The continuous phase (carrier fluid) consisted of a fluorocarbon oil HFE 7500 $\left(C_{3}F_{7}CF\left(OC_{2}H_{5}\right)CF{\left(CF_{3}\right)}_{2}\right)$ (3-ethoxy-dodecafluoro-2-tri-fluoro-methyl-hexane), which contained 2% wt. of a fluorosurfactant (dSURF), which was purchased from Fluigent (France). It is worth noting that fluorocarbon oil HFE 7500 is a highly chemically inert oil, purchased from Inventec (France), with a much higher mass density than water (1.61 g/cm${}^{3}$).

Although not the focus of the present study, we would like to mention here that the further condensation of droplets leads to the formation of highly monodisperse microcapsules, as shown in Figure 3 and described in detail in our previous study [26]. Briefly, to obtain microcapsules, droplets are first kept in the Petri dish at room temperature and atmospheric pressure for 48 h, and then the remaining oil and solvent are dried at 80 ${}^{{}^{\circ}}$C for 5 h to obtain fully condensed spherical zinc oxide microparticles [26].

To estimate the size of ZnO nanoparticles, we conducted high-magnification scanning electron microscopy (SEM) analysis on dried microcapsules, as shown in Figure 3c. SEM analysis enables us to give an approximate size of the ZnO nanoparticle building blocks of 100 nm.

After their production at the nozzle area of the microfluidic device, droplets were transported by the HFE 7500 carrier oil phase along the microfluidic channel and moved towards the outlet and a collecting Petri dish. Droplets were detected and analyzed in real time while flowing in the microfluidic channel by adding a fluorescent dye (rhodamine B in water at 0.1 mM) and using a home-built optofluidic setup, shown in Figure 2b and Figure 4.

## 3. Results and Discussion

### 3.1. First Experimental Evidence of the Formation of Dipolar Droplets

Figure 5 shows optical microscopy images of the organization at the oil/air interface of droplets prepared from a pure ZnO NP dispersion and from a mixture of ZnO NPs and rhodamine B (RhB) solution ([RhB] = 0.1 mM). It is worth noting that, at the used pH = 8.2, RhB molecules are in a zwitterionic form. These images show that, despite the addition of a small amount of RhB molecules (with respect to ZnO ‘units’), $\frac{\left[RhB\right]}{\left[ZnO\right]}=2\times 10^{-4}$, the resulting droplets organize into two different patterns: while pure ZnO microdroplets organize in a standard hexagonal close packing pattern, as seen in Figure 5a, RhB ZnO NP mixture droplets organize in an unexpected square lattice pattern, as shown in Figure 5b,c.

To the best of our knowledge, the existence of a long-range square order in a steady state at the oil/air interface with genuine liquid microdroplets is reported here for the first time. Such a square organization reveals the existence of both repulsive and attractive interactions between neighboring droplets, as illustrated in Figure 6.

Therefore, one should assume either (i) the formation and the coexistence of two types of droplets with homogeneously distributed electrical charges at their surfaces, as illustrated in Figure 7a,b, or (ii) the formation of ’dipolar droplets’ with a net electrical dipole, as illustrated in Figure 7c,d. The latter may result from the non-homogeneous distribution of ZnO NPs around the droplet interface.

We would like to mention that a square organization of droplets is reported in the literature, in a previous study by Wang et al. [32], where the droplet organization is controlled by the hydrodynamic flow inside the microfluidic channel. Our study, and the herein reported square organization, are different as they deal with a steady-state square organization of non-moving droplets, driven by the interaction and orientation of ZnO nanoparticles at the droplet interface.

Figure 7 shows different organization models of ZnO NP building units that may lead to the formation of surface-charged droplets and dipolar droplets at the droplet/oil and droplet/air interfaces. For the ‘surface-charged’ droplet model, one needs to assume the formation of two types of microdroplets with two opposite surface electrical charges: one with ZnO nanoparticles having, as a majority, their $Zn^{2+}$ planes oriented outwards (upwards) with regard to to the droplet interface—see Figure 7a—and the second (with a negatively charged surface) would be obtained according to the organization of ZnO nanoparticles with a ‘downwards’ orientation, as illustrated in Figure 7b.

To give a clearer view, we plot in Figure 8 the organization of ZnO nanoparticles for both ‘upwards’ and ‘downwards’ organizations, labeled ‘M1${}^{+}$’ and ‘M1${}^{-}$’, respectively. Obviously, for symmetry reasons, the global electric dipole of droplets with such orientations should be zero. It is mandatory that the two types of ‘upwards’ and ‘downwards’ orientations coexist in order to give rise to a square organization of droplets. Nevertheless, if interfacial tension is assumed to be strong enough to orient ZnO NPs in the same direction at the droplet interface, then this should lead also to only one type of orientation, either ‘upwards’ or ‘downwards’ orientation, which would lead in turn to a hexagonal organization of droplets. Hence, the ’surface-charged droplet’ model cannot explain the square organization and therefore should be rejected.

Another interesting feature of the organization of microdroplets shown in Figure 5c (and Appendix A) is that both square and hexagonal organizations may coexist in the same sample. In fact, such coexistence is also compatible with the dipolar feature of droplets, as illustrated in Figure 9.

Indeed, a square organization of microcapsules can be obtained with vertically oriented electrical dipoles but with an anti-parallel orientation (tail to head orientation), as sketched in Figure 9a, whereas a hexagonal organization of microdroplets in the same sample may be obtained with the same microcapsules if one changes alternatively the orientation of the electric dipoles of two adjacent raws with an angle of $+45^{\circ }$ and $-45^{\circ }$, respectively, as sketched in Figure 9b. In such a case, indeed, the interaction between the y-components (in-plane components) of the electric dipoles of microcapsules would be attractive, whereas the interaction between the z-components of electric dipoles would be repulsive. Nevertheless, since the z-components’ interaction is two-times smaller than the interaction between the y-components of the electric dipoles, the resulting hexagonal organization would be globally stable in this case.

### 3.2. Broken Spherical Symmetry of Droplets at the Oil/Air Interface

Essentially, because of the localisation of floating droplets at the oil/air interface, orientations of ZnO NPs at the upper interface (with air) and at the lower interface (with oil) may be different. We believe that such an asymmetric orientation of NPs could be at the origin of the breaking of the spherical symmetry of the ZnO NPs’ distribution around the droplet and the result of a global electric dipole, as illustrated in Figure 10. We consider first both a parallel and anti-parallel orientation (a-b) of ZnO NPs at the air/droplet interface and at the oil/droplet interface, respectively (Figure 10a,b). This type of configuration would be favored if the electric interactions are (i) weaker than the surface tension (with air) and (ii) stronger than the interfacial tension with oil (with dSurf molecules). On the other hand, we consider, in Figure 10c,d, an anti-parallel orientation of ZnO NP dipoles at the droplet/air interface and a parallel orientation at the droplet/oil interface, respectively. This second type of configuration would be favored if the electric interactions of ZnO NPs are (i) stronger than the surface tension (air) and (ii) weaker than the interfacial tension with oil (with dSurf molecules). It is worth noting that this balance between interfacial tension and the electrical interaction between ZnO NPs may be modified by the addition of electrical charges in the droplet’s inner phase (e.g., rhodamine B) or surfactant molecules in the oil phase (e.g., dSurf), as will be discussed later.

Based on energetic minimization arguments, at given experimental conditions, only one of the four configurations presented in Figure 10 should dominate. Let us suppose, for instance, that the main orientation is the one depicted in Figure 11a, where we consider three neighbouring droplets, labeled ‘1’, ‘2’, and ‘3’, with their electric dipoles all initially oriented upwards. Therefore, in order to minimize their electric energy, the droplet labeled ‘2’ should rotate in such a way that its electric dipole becomes oriented downwards, as illustrated in Figure 11b. However, since the rotation of droplet 2 would bring initially parallel-oriented NPs from the air/droplet interface to the oil/droplet interface, one would expect that the NPs would in turn change their orientation to an anti-parallel one in the oil phase (where dipolar interactions are assumed to be stronger than interfacial interactions). Simultaneously, and inversely, the same rotation of droplet 2 would bring initially anti-parallel NPs from oil to air (where interfacial interactions are assumed to be weaker) and hence one would expect that the droplets’ dipole orientation would change to a parallel one. Therefore, one would expect that the electric dipole of droplet 2 would return again to an upwards orientation state, as initially. Again, this model cannot explain alone the observed square organization of droplets.

### 3.3. Flow-Driven Droplet Electric Dipole Model (Fedd Model)

In order to better understand the organization of ZnO NPs at the droplet interface and to build a model that may better explain the organization of droplets in a square lattice (in the presence of rhodamine B molecules), we used a highly sensitive home-built optofluidic setup, shown in Figure 4 and described in more detail in a previous work [33]. It consists of a CW 532 nm laser source and a set of dichroic filters, mirrors, and a photo-multiplier tube (PMT), enabling us to detect and analyze, in a highly sensitive manner, the fluorescence signal emitted by the rhodamine B dye confined within flowing droplets.

Figure 12 shows a typical shape of the fluorescence intensity peak vs. time, which was collected from flowing droplets during their passage across the focused laser beam (maintained at a fixed position during the experiment). If we take into account the measured value of the droplets’ diameter (typically 62 μm) and the width of the fluorescence signal peaks (∼2 ms), we can deduce easily an average value of droplets’ velocity of approximately 3.2 cm/s. Interestingly, despite their relatively high velocity, droplets remain with a non-deformed spherical shape, as shown in Figure 12b. This feature indicates that the droplet interface becomes relatively rigid soon after being formed in the microfluidic channel. This feature represents further proof of the rapid adsorption of ZnO NPs at the droplet interface. Moreover, despite their perfect spherical shape and assuming *a priori* a homogeneous distribution of RhB molecules inside droplets, we remark that the fluorescence peaks exhibit a noticeably non-Gaussian, asymmetric shape, as clearly shown in Figure 12a.

A quick comparison of the integrated fluorescence intensities over the rear region (B) of the droplet and over its front region (A) shows that the fluorescence signal (which may be reasonably assumed to be proportional to the RhB molecules’ density) is approximately 1.5 times larger at the droplet’s back than at its front.

In fact, the observed asymmetric shape of the fluorescence peak may be interpreted easily based on the well-known tip-streaming effect in droplet microfluidics [27,28,29]. Indeed, it is well known that, in the presence of a surfactant, the recirculation flow induced by the motion of the droplet in the viscous carrier oil generates a heterogeneous distribution of the surfactant at the droplet interface: the surfactant’s interfacial density becomes higher at the rear region of the droplet than at its front, as illustrated in Figure 13a,b. Baret et al. [34] reported, for instance, a decade ago, direct evidence of the accumulation of surfactant molecules at the backs of droplets in the flow of a perfluorinated carrier oil. These authors used a home-made fluorescent non-ionic surfactant to monitor the build-up of the surfactant monolayer at the water/oil interface by the readout of the droplet interface fluorescence. In a more recent study, Hayat and El Abed [33] have shown that the accumulation of an ionic surfactant could be indirectly demonstrated by the accumulation of charged rhodamine B molecules at the back of the flowing droplets.

### 3.4. Effect of Addition of Rhodamine B Molecules

The observed transition from a hexagonal organization to a square organization of microdroplets at the micro scale indicates a change in the balance of ZnO nanoparticle interactions at the droplet interface when we add rhodamine B molecules. Indeed, the predominant electrical interactions between ZnO nanoparticles would favor anti-parallel (tail to head) orientations of the electrical dipoles of ZnO nanoparticles, whereas predominant interfacial interactions with the surrounding media (oil and air) would favor a parallel orientation of the electrical dipoles of ZnO nanoparticles. Consequently, one would expect that the screening of ZnO NPs’ electrical dipole interactions, by adding ions or charged molecules such as rhodamine B, would lead to a shift in the balance of interactions between nanoparticles and the determination of the final organization of ZnO nanoparticles at the droplet interface.

Moreover, our results shown in Figure 12a provide direct evidence of the existence of a strong interaction between dSurf surfactant molecules and rhodamine B molecules. If this was not the case, the intensity of the droplet fluorescence would have been more or less of a Gaussian type.

Because of the demonstrated accumulation of RhB molecules at the rear part of the droplet/oil interface and the electric charges carried by RhB molecules, we suggest that the higher density of RhB molecules at the rear parts of the droplets enhances the screening of repulsive electric interactions between parallel-oriented electric dipoles, enhancing therefore the parallel orientation of ZnO NPs at the rear part of the droplet interface, as illustrated in Figure 13c. Inversely, because of the depletion of surfactant and RhB molecules at the front parts of the flowing droplets, electric interactions between ZnO nanoparticles are less screened and a tail-to-head (anti-parallel) orientation should be favored, as illustrated in Figure 13d.

Finally, we would like to mention that the observed square organization of droplets has long-term stability that can last several days (or weeks, depending on the storage conditions). These results show unambiguously that the parallel orientation of ZnO NPs is not a transitory process. It persists even after the evaporation of solvent and oil and the transformation of droplets into microcapsules, as will be detailed in a following study.

## 4. Conclusions

We investigated, in this work, using droplet microfluidics, optical microscopy, and high-throughput fluorescence optofluidic detection, the organization of ZnO nanoparticles at the very early stage of ZnO droplet formation, and we show for the first time a square lattice organization of liquid microdroplets. Our results show unambiguously the dipolar self-organization of ZnO nanocrystals at the surfaces of the microdroplets at the early stage of droplet formation. We suggest that the rise of the global electric dipole moment of droplets is triggered by the microfluidic flow and the well-known tip-streaming effect in droplet microfluidics.

## Figures and Tables

**Figure 1 micromachines-13-01522-f001:**
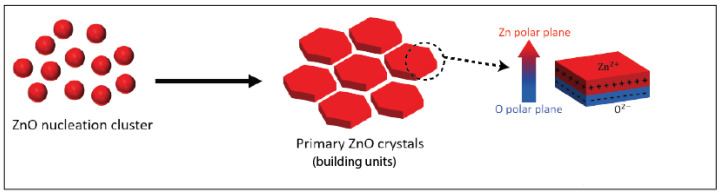
Nucleation and growth scheme of ZnO nanoparticle building units.

**Figure 2 micromachines-13-01522-f002:**
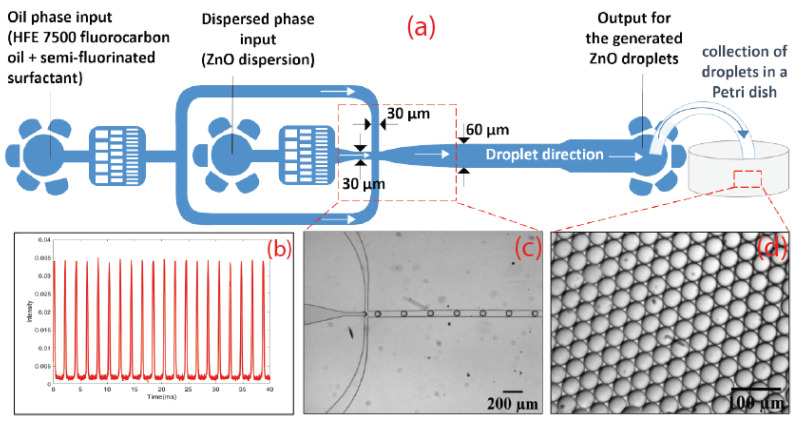
(**a**): Schematic illustration of the flow-focusing microfluidic design used for the production of zinc oxide droplets (and microcapsules); (**b**) real-time fluorescence detection of flowing droplets using rhodamine B dye and our optofluidic setup; (**c**,**d**): optical micrographs of monodisperse zinc oxide droplets in the microfluidic channel and in the Petri dish, respectively.

**Figure 3 micromachines-13-01522-f003:**
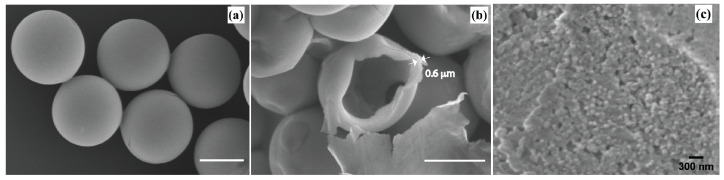
(**a**) SEM images of 16.5 μm zinc oxide microcapsules obtained from 62 μm droplets; (**b**) shell thickness was found to be approximately 0.6 μm; (**c**) high-magnification SEM on shell microcapsules showing the size of ZnO NPs building blocks of approximately 100 nm. Scale bars: (**a**,**b**) 10 μm, (**c**) 0.3 μm.

**Figure 4 micromachines-13-01522-f004:**
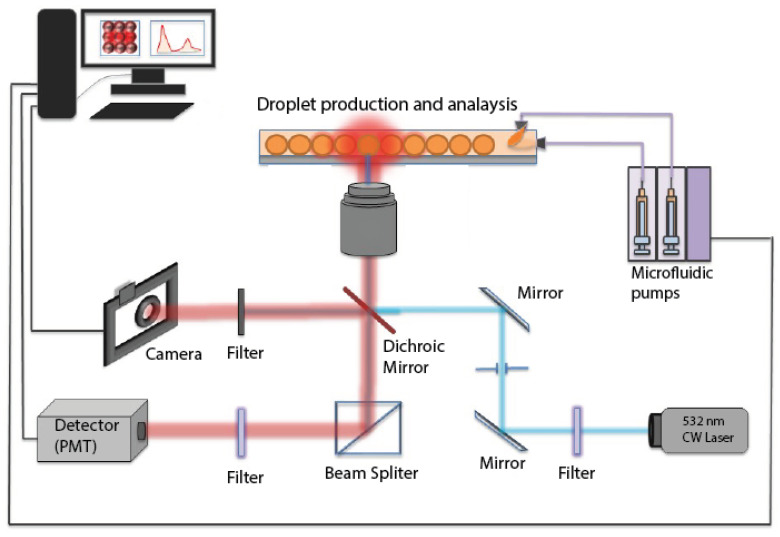
Optofluidic setup used for real-time fluorescence measurements recorded from flowing droplets in the microfluidic channel.

**Figure 5 micromachines-13-01522-f005:**
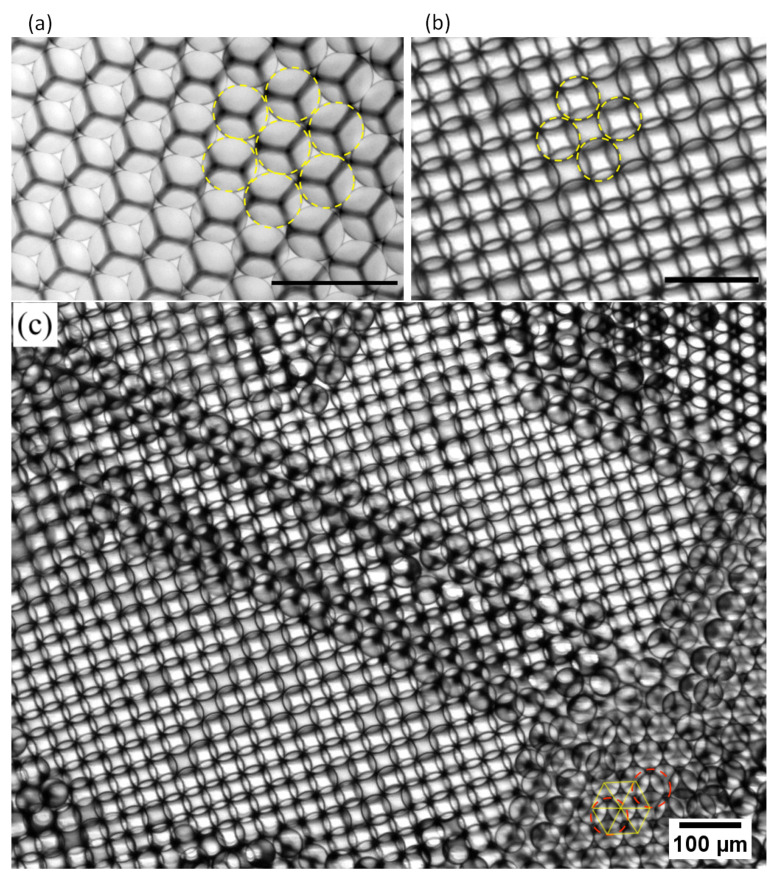
Optical microscopy images of two layers of ZnO NP droplets collected at the oil/air interface (in a Petri dish), showing standard hexagonal packing in the absence of rhodamine B dye (**a**) and an unexpected square organization when rhodamine B dye is added (**b**,**c**). This square organization highlights the existence of spontaneous electrical polarization within these droplets. Hexagonal organization may coexist with square organization, as seen in the lower right of the image shown in (**c**); see text for explanation. Scale bars: 100 μm. See also Appendix A for the visualization of the dynamics of the observed square organization of droplets.

**Figure 6 micromachines-13-01522-f006:**
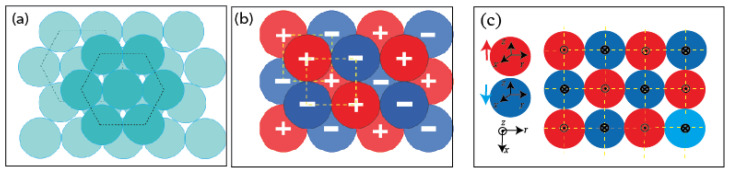
Schematic illustration of ZnO microdroplet organizations in hexagonal (**a**) and square (**b**,**c**) lattices. In (**c**), the ‘·’ dot sign surrounded by a circle in red spheres represents an electric dipole oriented upwards (along the direction of the z-axis), whereas the ‘×’ cross sign surrounded by a circle in blue spheres represents a downward-oriented electric dipole (along the opposite direction of the z-axis).

**Figure 7 micromachines-13-01522-f007:**
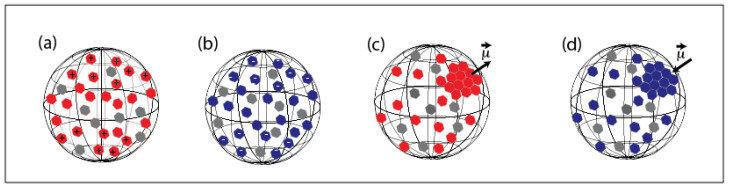
(**a**) Positive and (**b**) negative surface-charged models of droplets for which NPs are homogeneously distributed around the droplet, unlike in dipolar droplet models (**c**,**d**); grey color is used for NPs with either a positive surface charge (upwards orientation) or with a negative charge (downwards orientation). For each model, the majority of NPs are oriented either upwards or downwards, in a such way that a global surface charge is obtained: (a)→(+), (b)→(−), (c)→(+) and (d)→(−).

**Figure 8 micromachines-13-01522-f008:**
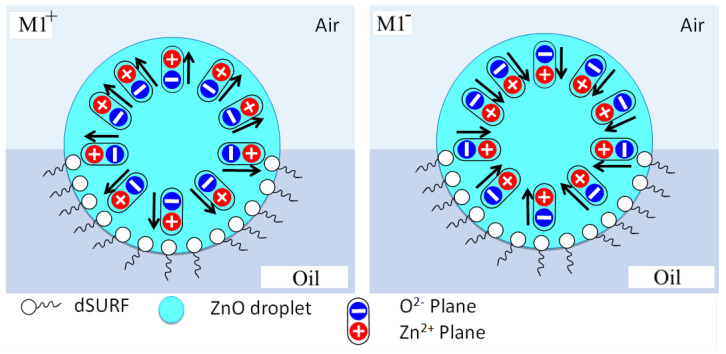
Schematic illustration of ZnO NP organizations leading to positively charged droplets (M1${}^{+}$, **left**) and negatively charged droplets (M1${}^{-}$, **right**) charged droplets, respectively. The coexistence of both organizations is mandatory to explain the observed square lattice of droplets.

**Figure 9 micromachines-13-01522-f009:**
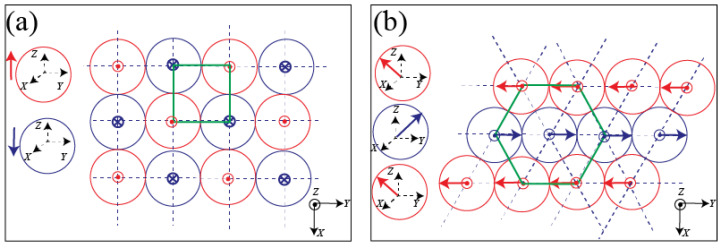
Suggested models for the orientation of electric dipoles of microdroplets with a square (**a**) and a hexagonal (**b**) lattice organization: the tilting of the direction of electric dipoles of droplets by an angle of $+45^{\circ }$ and $-45^{\circ }$, respectively, may explain the transition from a square organization to a hexagonal organization and hence the coexistence of both types of organization of droplets.

**Figure 10 micromachines-13-01522-f010:**
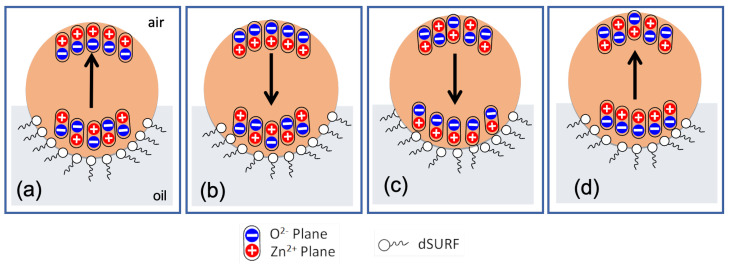
Schematic illustration of different possible asymmetric orientations of ZnO nanoparticles at droplet/air and droplet/oil interfaces (leading to dipolar microdroplets); (**a**,**b**): parallel orientation of ZnO NPs at the droplet/air interface and an anti-parallel orientation at the droplet/oil interface; (**c**,**d**): parallel orientation of ZnO NP dipoles at the droplet/oil interface and an anti-parallel orientation at the droplet/air interface.

**Figure 11 micromachines-13-01522-f011:**
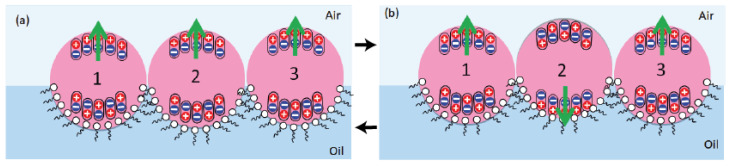
Schematic illustration of dipolar interactions between dipolar droplets. Electric bipoles oriented: (**a**) upwards and (**b**) downwards.

**Figure 12 micromachines-13-01522-f012:**
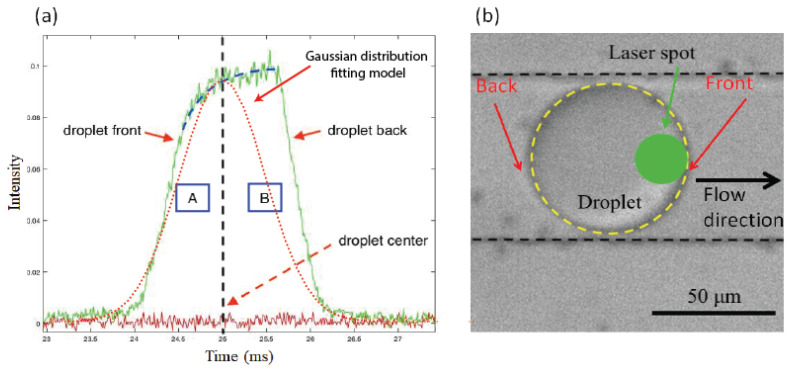
Shape of the fluorescence intensity signal versus time emitted from RhB-doped ZnO microdroplets in the presence of dSURF surfactant (**a**). The corresponding optical microscopic image of the microdroplet flowing inside the microfluidic channel (**b**).

**Figure 13 micromachines-13-01522-f013:**
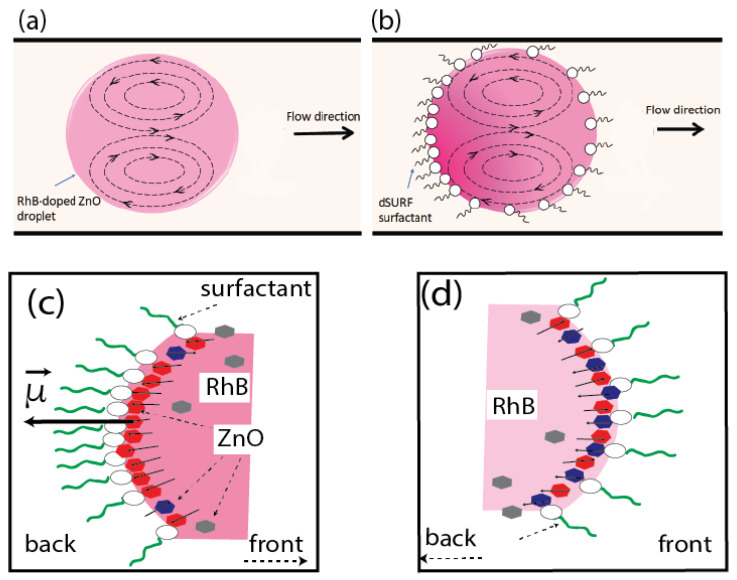
Schematic illustration of tip-streaming effect in droplet carried by oil flow: (**a**) without surfactant and in the presence of surfactant (**b**); (**c**,**d**): illustration of non-homogeneous distribution of ZnO NPs in the back area (**c**) and the front area (**d**) of the droplet; red hexagons represent ZnO NPs with their electric dipoles oriented outwards and blue ones represent those with electric dipoles oriented inwards.

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
