# Peer review of "Hierarchical Self-Assembly of Dipolar ZnO Nanoparticles and Microdroplets"

_micromachines, 2022, doi:10.3390/mi13091522_

Round 1

Author Response

Authors responses to Reviewer 1 comments and criticisms on Ghifari et al. manuscript submitted to Micromachines on July 29 July 2022 under n° 1865929

--------------------------------------------------

Najla Ghifari et al. have reported the self-assembly of dipolar nanoparticles and microdroplets into different configurations, which reply on the electrostatic force interaction and the tip-steaming effect. There shows some experimental data and assumptions. However, I do not encourage to publish the work in this version, due to the shortage of solid data to demonstrate the charges playing a critical role for the droplets organization. Without any electrostatic forces, square pattern can also be obtained, please see for example Juan Wang et al. Microfluidic Induced Controllable Microdroplets Assembly in Confined Channels, 2015.

Response to the general conclusion of the reviewer: We would like first to thank the reviewer for his (or her) overall nice evaluation of our manuscript and comments. We respond in the following to all his (her) cristisms. But as a response for his (her) general comment, we would like to emphasize that actually, the main goal of this work was firstly to demonstrate experimentally that a square organisation of settle droplets may be obtained at the oil/air interface, as supported with our data. Secondly, we present a model, based on tip-streaming effect, which enables to explain the rise of a global electric dipole at the droplet scale based on nanoparticles organisation. The reviewer mentions a previous study of Wang et al. on flowing droplets in a microchannel, where a square organisation of droplets is observed. We would like to point out that square organisation reported by Wang et al. is controlled by the flow and the hydrodynamics and not by nanoparticles or molecular interactions. Our study is different, it deals with non-moving droplets (zero velocity) which organize spontaneously in a square lattice at the oil/air interface driven by the orientation of nanoparticles at the droplet interface. Therefore, the general reviewer criticism about the need to investigate the effect of flowrates on droplets organisation is not relevant for this study. We added a paragraph about the work of Wang et al. and the related reference in bibliography:

We would like to mention that a square organisation of droplets was previously reported by Wang et al. \cite {wang2015microfluidic}, where droplets organisation is controlled by the flow and the hydrodynamics. This study deals with non-moving droplets which organize spontaneously in a square lattice at the oil/air interface driven by the orientation of nanoparticles at the droplet interface.”

  1. In experimental section, you mentioned that ‘Droplets were kept in the Petri-dish at room temperature and atmospheric pressure during 48 hours, then the remaining oil and solvent were dried at 80 ◦C for 5 hours to obtain fully condensed 82 spherical zinc oxide microparticles, as reported in our previous study [26]’, so when did you observe the organized configurations by droplets? I mean you observe the fresh microdroplets collected in a petri-dish once the produced droplets flows in or you wait quite a long time till the droplets covered the entire petri-dish and evaporate them at 80 degree, then transfer for observation?

Response: We mentioned in the introduction of our manuscript, that we focus in the present work on the organisation of ZnO NPs and their droplets in the droplet state. Therefore, all observations are done either with just produced droplets while flowing in the microfluidic channel (for fluorescence measurements) or freshly collected droplets in the Petri-dish. We start analysis soon after we collect enough droplets to form monolayers and bilayers of droplets at the oil/air interface, which cover fully or a large part of the surface of the Petri-dish.

The last paragraph we added in the experimental protocol we followed to produce microcapsules, described in our previous study [26]. To avoid any confusion, we modified this paragraph in the revised manuscript.

  1. I strongly suggest to control the flow rate to observe the organization configurations as the volume faction of droplets filled in petri-dish (how large you used?) is affected by the droplets production frequency, which turns out to affect the patterning configurations.

Response: We thank the reviewer for this suggestion and will consider to carry such investigation in a future work. Indeed, as mentioned above, we focused in the present study on the organisation of microdroplets in a steady medium, at the oil/droplet interface, unlike in the study mentioned by the reviewer (Wang et al.), which is added in the references list of the revised version. Wang et al. article reports on the organisation of droplets carried by a hydrodynamic flow in a microfluidic channel (and not in a steady state), where the ordering is induced by the hydrodynamic flow itself and not by intrinsic electrostatic interaction between nanoparticles as shown in our work. 

  1. Could the authors please provide the proof that the capsulated ZnO nanoparticles locating at the droplet surface?

Response: As mentioned in 2nd paragraph of page 8 and shown in Fig. 10 (b), despite the relatively high velocity of the flow, droplets remain un-deformed with a perfect spherical shape. This feature indicates that the droplet interface becomes relatively rigid soon after their formation in the microfluidic channel, which we attribute to a rapid adsorption of ZnO NPs at the droplet interface. Nevertheless, in our model, it’s not mandatory that ALL NPs adsorb on the surface of the droplet. Only an adsorbed fraction of ZnO NPs at the droplet interface may be responsible of the rise of a global electric dipole. NPs located in the bulk of droplet have no effect as they should be randomly oriented inside the droplet. Droplets without ZnO nanoparticles encapsulated didn’t show any square organisation, even in the presence of RhB and surfactant molecules.

  1. In figure 4c, except the presence of square patterns, do authors have any explanations on the other patterns, for example at top right and bottom right part?

Response: we thank the reviewer for this remark. Indeed, we can observe in figure 4c, the presence of few hexagonally organised of droplets, as seen in the top right and bottom right of figure 4c. We modified this figure, where we emphasized the hexagon (in yellow) formed by centres of the six droplets involved in this structure, where we can notice the presence of two droplets (enhanced with red dashed circles) on the top of the hexagonal structure. This is a kind of partial 3D rhombohedral structure and not a pure 2D hexagonal structure. We believe that these upper droplets stabilize the observed hexagonal organisation of droplets with anti-parallel oriented dipoles.

  1. Could the authors quantify the forces among electrostatic interaction, surface tension and interface tension instead of through assumption to predict the configurations? (see our response below).
  2. It is still not very clear how does the addition of R6G affect the square lattice formation? Could the authors please give more explanations on how does R6G screen the ZnO particles? And Why does the ZnO particles orientate parallely at the rear side of droplets, while orientate anti-parallely at the front side of droplets?

Responses (to 5 and 6): The effect of RhB (we did not use R6G dye) on the type of obtained organisation of droplets (from hexagonal to square organisation) is clearly evidenced experimentally in this study. This is the main topic of this study. A theoretical study involving calculations of dipolar interactions and their screening with electrical charges (including RhB’s) is under investigation and results will be submitted later for publication. Actually, we carried out experiments, where we add ions in the dispersed phase (KCl= 0.1 mM) and observe a similar effect than with RhB. We believe, as mentioned in the first paragraph of section 3.4 (page 9), that if predominant electrical interactions between ZnO nanoparticles may be weakened by the screening of electrical charges (brought by the charged RhB molecules), then the organisation of NPs would change from an anti-parallel (tail to head) orientation to a parallel orientation, due the surface tension, which would orient NPs in the same direction by regards to the interface. Because of the spherical symmetry of droplets, such re-orientation of NPs from anti-parallel to parallel is not efficient alone to explain the appearing of a net of electrical dipole.  One needs to admit a breaking of the spherical symmetry by assuming a non-homogeneous polar distribution of NPs around the surface of droplets. This polar distribution is induced by tip-streaming effect acting on the flowing droplets.

  1. Please carefully check the grammars and spellings.

Reviewer 2 Report

The article is excellent from a scientific point of view, but the researchers did not mention the crystal size of the prepared ZnO NP. Also, there is no examination that proves that the prepared ZnO NP is nanoparticles.

Author Response

Reviewer 2: The article is excellent from a scientific point of view, but the researchers did not mention the crystal size of the prepared ZnO NP. Also, there is no examination that proves that the prepared ZnO NP is nanoparticles.

Response: We sincerely thank the reviewer for the very nice comments and very good appreciation of our work. We added a paragraph in page 3 and a figure (figure 3) showing high resolution scanning electronic microscopy image which shows clearly the nanoparticles feature of ZnO building blocks and a size of about 100 nm for such nanoparticles.

Round 2

Reviewer 1 Report

I still have some doubts on the explanation of square lattice observation driven by the electrostatic  force.

for example:

1. In experimental section, you mentioned that ‘Droplets were kept in the Petri-dish at room temperature and atmospheric pressure during 48 hours, then the remaining oil and solvent were dried at 80 ◦C for 5 hours to obtain fully condensed 82 spherical zinc oxide microparticles, as reported in our previous study [26]’, so when did you observe the organized configurations by droplets? I mean you observe the fresh microdroplets collected in a petri-dish once the produced droplets flows in or you wait quite a long time till the droplets covered the entire petri-dish and evaporate them at 80 degree, then transfer for observation?

Response: We mentioned in the introduction of our manuscript, that we focus in the present work on the organisation of ZnO NPs and their droplets in the droplet state. Therefore, all observations are done either with just produced droplets while flowing in the microfluidic channel (for fluorescence measurements) or freshly collected droplets in the Petri-dish. We start analysis soon after we collect enough droplets to form monolayers and bilayers of droplets at the oil/air interface, which cover fully or a large part of the surface of the Petri-dish.

---Yes, this is exactly my concerns. If the square lattice caused by the charges of ZnO particles, then I expect you could observe square lattice within very a few droplets, not necessary to obtain a monolayer or even more for observation. Do you have any images/videos of droplets organization once they produced and flows into the petri-dish to see how do they arrange in real-time?

Droplets without ZnO nanoparticles encapsulated didn’t show any square organisation, even in the presence of RhB and surfactant molecules.

---Regarding to this, do the authors have any videos to show, for example the dynamic organization process?

Author Response

Dear reviewer, we understand your doubts, as the observation of a square organisation of droplets in a settle medium is indeed puzzling and unexpected (we had the same feeling too at the beginning). Nevertheless, our experimental results show that this happens. We added, as you requested, a video with a time counter in seconds and minutes inserted, which shows the dynamics of organisation and re-organisation of droplets in a square pattern, at mn time scale, after we have shaky gently the droplets at the surface of oil in the Petri-dish. We supply the video as a Supplementary Material in the revised version of the manuscript). Please, have a close look to the line of (darker) droplets in the rectangular set of droplets which re-organise during time in a such a way to insert fully into the large square lattice (ends at a time around 22 seconds). It This shows that square organisation of droplets at this time scale is stable and not just a transient state. We hope we convinced you this time that the interactions behind such organisation are intrinsic physical forces. Please let us know if you have any suggestion about any other physical interaction than electrostatic ones.

Round 3

Reviewer 1 Report

Very sorry, I am still not very confident from what I see from the current video as thousands of droplets organized there. That is to say there are much more forces involved. So could the author provide the video showing the whole organization process, particularly from a few (less than 10 droplets ) up to a monolayer? Shouldn't we observe the square lattice driven by electrostatic force with a very small amount of droplets?

Author Response

Dear Reviewer,

We are also very sorry that we could not convince you despite our efforts to do so and our responses to all your previous requests.

To try again to convince you, we added another video as you requested. Unfortunately, it’s very difficult to start with few droplets (as you request), since the production rate of droplets  is several (~500) hundreds of droplets/second. Therefore, it’s not possible to follow the organisation of droplets from few droplets to thousands. Moreover, droplets aggregate together in the tubing before they are released and organize at the surface of the Petri-dish. Nevertheless, we hope that you will appreciate our additional effort to meet your criticism by adding the new video, as a supplementary material (together with the previous video with thousands of droplets).

Actually, we never claimed that only square organisation of droplets is observed. On the contrary, we modified our manuscript in order to emphasize that hexagonal organisation may be observed. We modified figure 5 accordingly. We add now, in the new revised manuscript, an additional paragraph and a new figure (figure 9) to explain how a hexagonal organisation may be obtained with dipolar droplets. Obviously, a hexagonal organisation of droplets may be obtained with non-dipolar droplets but only dipolar droplets can explain the observation of a square organisation (even if they are not in majority within the sample) or its coexistence with a hexagonal organisation.

Another interesting feature of the organisation of microdroplets shown in Fig. 5(c) (and supplementary material videos) is that both square and hexagonal organisations may coexist in the same sample. Actually, such a coexistence is also compatible with the dipolar feature of droplets, as illustrated in figure 9.

Indeed, square organisation of microcapsules can be obtained with vertically oriented electrical dipoles but with an anti-parallel orientation (tail to head orientation), as sketched in figure 9(a), whereas hexagonal organisation of microcapsules in the same sample may be obtained with the same microcapsules if one changes alternatively the orientation of the electric dipoles of two adjacent raws with an angle of +45 and −45, respectively, as sketched in figure 9(b). In such a case indeed, the interaction between y-components (in-plane components) of the electric dipoles of microcapsules would be attractive whereas the interaction between z-components of electric dipoles would be repulsive. Nevertheless, since the z-components interaction is two times smaller than the interaction between the y-components of the electric dipoles, the resulting hexagonal organisation would be globally stable in this case.”

Figure 9. Suggested models for the orientation of electric dipoles of microdroplets with a square (a) and a hexagonal (b) lattice organisations: the tilting of the direction of electric dipoles of droplets by an angle of +45◦ and −45◦, respectively, may explain the transition from a square organisation to a hexagonal organisation and hence the coexistence of both types of organisation of droplets.
